# Deciphering the Enigma of the Histone H2A.Z-1/H2A.Z-2 Isoforms: Novel Insights and Remaining Questions

**DOI:** 10.3390/cells9051167

**Published:** 2020-05-08

**Authors:** Manjinder S. Cheema, Katrina V. Good, Bohyun Kim, Heddy Soufari, Connor O’Sullivan, Melissa E. Freeman, Gilda Stefanelli, Ciro Rivera Casas, Kristine E. Zengeler, Andrew J. Kennedy, Jose Maria Eirin Lopez, Perry L. Howard, Iva B. Zovkic, Jeffrey Shabanowitz, Deanna D. Dryhurst, Donald F. Hunt, Cameron D. Mackereth, Juan Ausió

**Affiliations:** 1Department of Biochemistry and Microbiology, University of Victoria, Victoria, BC V8W 3P6, Canada; mscheema@uvic.ca (M.S.C.); katrina.vanessagood@gmail.com (K.V.G.); Cindy.Kim@smus.ca (B.K.); connoro@uvic.ca (C.O.); melissfreeman@outlook.com (M.E.F.); phoward@uvic.ca (P.L.H.); ddryhurst@gmail.com (D.D.D.); 2Institut Européen de Chimie et Biologie, Univ. Bordeaux, 2 rue Robert Escarpit, F-33607 Pessac, France; heddy.soufari@u-bordeaux.fr (H.S.); cdmackereth@gmail.com (C.D.M.); 3Inserm U1212, CNRS UMR 5320, ARNA Laboratory, Univ. Bordeaux, 146 rue Léo Saignat, F-33076 Bordeaux, France; 4Department of Neurosciences & Mental Health, the Hospital for Sick Children, Toronto, ON M5G 1X8, Canada; gildastefanelli@gmail.com (G.S.); iva.zovkic@utoronto.ca (I.B.Z.); 5Environmental Epigenetics Group, Department of Biological Sciences, Florida International UniversityNorth Miami, FL 33181, USA; cirorc80@gmail.com (C.R.C.); jeirinlo@fiu.edu (J.M.E.L.); 6Department of Chemistry and Biochemistry, Bates College, 2 Andrews Road, Lewiston, ME 04240, USA; kzengele@bates.edu (K.E.Z.); akennedy@bates.edu (A.J.K.); 7Department of Psychology, University of Toronto Mississauga, Mississauga, ON L5L 1C6, Canada; 8Department of Chemistry, University of Virginia, Charlottesville, VA 22904, USA; js4c@virginia.edu (J.S.); dfh@virginia.edu (D.F.H.); 9Department of Pathology, University of Virginia, Charlottesville, VA 22903, USA

**Keywords:** replication dependent (RD) and replication independent (RI) histone variants, H2A.Z-1 and H2A.Z-2, evolution, NMR, transcription, spermatogenesis, development

## Abstract

The replication independent (RI) histone H2A.Z is one of the more extensively studied variant members of the core histone H2A family, which consists of many replication dependent (RD) members. The protein has been shown to be indispensable for survival, and involved in multiple roles from DNA damage to chromosome segregation, replication, and transcription. However, its functional involvement in gene expression is controversial. Moreover, the variant in several groups of metazoan organisms consists of two main isoforms (H2A.Z-1 and H2A.Z-2) that differ in a few (3–6) amino acids. They comprise the main topic of this review, starting from the events that led to their identification, what is currently known about them, followed by further experimental, structural, and functional insight into their roles. Despite their structural differences, a direct correlation to their functional variability remains enigmatic. As all of this is being elucidated, it appears that a strong functional involvement of isoform variability may be connected to development.

## 1. Introduction

### 1.1. Preamble

Histones represent the most abundant group of chromosomal proteins. From the structural point of view they can be grouped into core histones and linker histones [1]. Core histones (H2A, H2B, H3 and H4) consist of a highly specific dimerizing core histone fold domain (HFD) [2]. They are responsible for the formation of the protein core around which DNA wraps into a left-handed super-helical organization, eventually leading to the fundamental subunit of chromatin known as the nucleosome. Linker histones (histones of the H1 family), as their name indicates, bind to the linker DNA connecting adjacent nucleosomes in the chromatin fiber. Histones can also be functionally classified into replication dependent (RD) (expressed during replication) and replication independent (RI) (expressed throughout the cell cycle). The messenger RNAs (mRNAs) of RI histones are polyadenylated at their 3’ end and usually contain introns; in contrast, the mRNAs of RD histones contain a highly specific stem-loop at their 3’ end and do not contain introns [3]. RD histone genes in metazoans are present in large copy numbers as they constitute most of the histones incorporated in the de novo assembly of chromatin during DNA replication [4]. RI histones instead are either present as a single copy or in some instances may contain a restricted number of isoforms, as is the case of the histone variant H2A.Z.

This paper presents a combination of a literature review on the two main RI H2.A.Z histone isoforms, H2A.Z-1 and H2A.Z-2, as well as some new experimental data. As such, it will start with an introduction briefly reviewing the history of the two isoforms, followed by what is currently known about their enigmatic structure and function. The Introduction will then be followed by a Materials and Methods section pertaining to some new experimental data that will be described in the Results section, followed by an overall joint discussion.

### 1.2. Histone H2A.Z in Brief

The first identification of replication independent (RI)-H2A.Z took place in the early 1980s in the lab of Dr. Bill Bonner which would correspond to what we know today as H2A.Z-1 [5,6]. By about the same time, cDNA was synthesized from chicken embryo in Julian Wells’ lab in Australia, encoding a histone which was named H2A.F and which matches the sequence of H2A.Z-2 [7]. However, the relation of the chicken H2A.F to H2A.Z was at the time uncertain and the connection between the two was not established [7].

Histone H2A.Z protein levels are estimated to comprise approximately 5% of the total H2A histones in vertebrates [6]. For many years, histone H2A.Z was assumed to be a single protein RI histone variant with different names in different organisms and hence its initial functional attributes were generically referred to this assumption. As such, one of the most enigmatic characteristics of histone variant H2A.Z has been its generic association to both transcriptionally active and repressive chromatin [8]. Partially addressing the problem in *Saccharomyces cerevissiae* (yeast), where H2A.Z is present in only one copy known as Htz1, it was shown that H2A.Z had antisilencing activity at the boundaries between euchromatin and heterochromatin [9]. In a recently published review on this variant, it was suggested that such a seemingly contradictory transcriptional role is present to buffer phenotypic noise which ultimately modulates the efficiency of transcription [10]. Similarly, it had been proposed earlier that H2A.Z mediates such contrasting activities by acting as a general facilitator that generates access to both activating and repressive complexes [11]. It also may be that, as pointed out by Giaimo and co-authors [10], posttranslational modifications (i.e., acetylation and ubiquitination) are at the core of the controversial H2A.Z involvement in the regulation of gene expression [10,12,13,14].

It was also shown that this histone variant, which in *Drosophila* is called H2Av and exists as a single copy gene encoding a structural and functional hybrid between H2A.X and H2A.Z, was indispensable for survival [15,16,17]. Thus, H2A.Z importantly became the only histone variant whose lack of expression is lethal for the organism. Although mice knock out experiments demonstrated that H2A.Z is required for early mammalian development, its complete indispensability was not as clear cut. This is probably due to the fact that only the gene for one of the two H2A.Z vertebrate isoforms was knocked out [18]. It is likely that abolition of the two isoforms in vertebrates (see below) would be completely lethal. Interestingly, the single copy gene for yeast, Htz1, is not necessary for its survival. This however might be explained by the fact that in metazoan organisms, where both H2A.Z and H2A.X occur in similarly low percentages compared to their replication dependent (RD) H2A counterpart, both RI variants are important for different aspects of DNA repair and genome integrity [19,20,21,22,23,24]. Yet, in yeast, H2A.X completely replaces RD H2A, and deficiency of Htz1, which is also present in small amounts, might be partially functionally accounted for by the massive presence of H2A.X in the chromatin of this organism.

In plants, the H2A.Z variant represents a critical player in regulating environmental responses [25,26,27] as well as participating in other processes including DNA repair, somatic homologous recombination, and replication origin specification [28].

H2A.Z plays a role in the modulation of gene expression, although its effects in this regard are not completely understood. It has been proposed that high levels of H2A.Z cover the body of inactive genes involved in environmental responses [29], while the presence of H2A.Z in +1 nucleosomes usually correlates with active genes [27]. However, there are contradictory reports in this later case that could be explained to some extent by the presence or absence of specific post-translational modifications [30] adding another layer of complexity to the function of H2A.Z in gene expression. In addition, in *A. thaliana*, as in the case of animals, regions of DNA methylation are quantitatively deficient in H2A.Z, as seen at sites of DNA methylation in the bodies of actively transcribed genes and in methylated transposons [31]. This observation could be indicative of DNA methylation influencing chromatin structure and gene silencing by excluding H2A.Z. This is further supported by chromatin immunoprecipitation assays revealing an inverse correlation between gene transcript levels and H2A.Z deposition [32]. Overall, it seems that H2A.Z variants do not participate directly in the activation of transcription, but rather they commit a chromatin state competent for activation by other factors or to protect from silencing by DNA methylation [33]. An epigenetic role for plant H2A.Z is hinted by the immunolocalization of this variant associated with chromosomes during mitosis [32], potentially promoting a rapid reactivation of genes in response to specific signals.

Aside from the transcriptional and double stranded break DNA repair implications, this histone variant is also not surprisingly involved in chromosome segregation and in a variety of other functions such as memory consolidation and more recently in chromatin licensing and activation of early origins of replication in conjunction with H4K20me2 [34,35,36]. For more comprehensive information about H2A.Z’s functional roles, the reader is directed to [10].

At the structural level, histone H2A.Z consists of a typical HFD flanked by unfolded N- and C-terminal domains (tails) [2]. Hence, it interacts with the rest of the core histones to form a protein core octamer that serves as a scaffold for the assembly with DNA into nucleosomes in a very similar way as RD H2A [37]. The crystallographic structure of the nucleosome consisting of two H2A.Z-H2B dimers with the H3-H4 tetramer indicates that the loop L1 region between alpha helix 1 and alpha helix 2 of the HFD is in a slightly different organization compared to the nucleosome consisting of RD H2A [38]. This is the region where the two H2A.Z proteins of the histone core hold together the two superhelical gyres of DNA in the nucleosome. The N-terminal unstructured regions of RD H2A, and probably those of H2A.Z, are also sites of interaction with DNA in the nucleosome [39]. Interestingly, the two H2A.Z isoforms have differing amino acids (T/A 14) and (S/V 38) (Figure 1C) within these two respective regions (Figure 2A and Figure 3A), as will be discussed later.

### 1.3. Rediscovery of the Two H2A.Z Isoforms

By the end of 2003, an exhaustive effort to isolate and purify the native RI H2A.Z from chicken liver histones revealed that the purified protein exhibited two electrophoretic bands in sodium dodecyl sulfate polyacrylamide gel electrophoresis (SDS-PAGE) (Figure 1A). However, whether this was the result of post-translational modification or an unknown degradation due to the long chromatographic purification steps remained a mystery. The problem was not resolved until 2005 when the laboratory of Donald Hunt in the Department of Chemistry at the University of Virginia, using a novel sequential ion–ion reaction and top-down tandem mass spectrometry (MS-MS) approach developed in this lab found that the two bands were the result of two proteins differing in their amino acid sequences (Figure 1B). This difference in the case of chickens affects four amino acids: A12T, A14T, T38S, A127V, from H2A.Z-2 to -1, respectively, and in mammals it involves only the last three of these amino acid changes (Figure 1C). The H2A.Z-1 and H2A.Z-2 isoforms are encoded by two different genes: *H2AFZ* and *H2AFV* located on chromosomes 4q23 and 7p13 respectively in humans [4,40].

### 1.4. What We Currently Know

The presence of two main H2A.Z isoforms is not restricted to vertebrates, and while in some invertebrates there appears to be only one isoform present, in many others the appearance of two isoforms seems to have been a recurrent phenomenon in the course of evolution (Figure 2) [18,42]. As shown in this figure, a unique isoform is also present in plants. In this kingdom, all H2A.Z variants cluster together as a monophyletic group in the evolutionary tree of H2A, suggesting that H2A.Z evolved directly and once from the ancestral H2A [40]. Similarly, the sequence conservation of H2A.Z among different plant species is considerably higher than that of other H2A variants [29]. In contrast to deuterostome animals, plant histone H2A.Z does not display an evolutionary differentiation into subtypes H2A.Z-1 and H2A.Z-2 [40]. Instead, three functional genes encoding H2A.Z, *HISTONE H2A 8* (*HTA8*), *HTA9*, and *HTA11* can be found in *Arabidopsis thaliana* [33]. Among them, only the *HTA9* gene is expressed independently of the cell cycle, as would be expected of a histone variant, possibly suggesting that not all *Arabidopsis* H2A.Z variants have redundant functions. However, single knockout experiments still suggest a certain level of redundancy between *HTA9* and *HTA11* [25,43]. Nonetheless, it has been shown that *HTA9* and *HTA11* mutants do not compensate for the loss of these proteins, therefore attributing specialized functions to the H2A.Z variant in *Arabidopsis* [27].

The identification of two H2A.Z isoforms in animals allows for a better definition of their function. Their individual functions, when taken together, amount to most of the functionality described for H2A.Z in the more ancestral organisms that contain only one isoform (i.e., yeast), strongly suggesting that gene duplication led to the subfunctionalization of H2A.Z [44,45,46].

At present, it is still not clear whether the amino acid differences between the two isoforms (Figure 1C) confer structural differences affecting their functions. This could be by altering their interaction with the rest of the core histones and/or the DNA or by affecting the affinity of their interaction with their chaperones. While a good amount of evidence has been gathered for the former, only one report so far has indicated a preferential interaction of H2A.Z-1 with Bromodomain-containing protein 2 (Brd2) [13,47].

In what follows, we will describe the functions that have been ascribed to each of the isoforms so far and their significance.

Following the identification of the RD H2A.Z-1 and H2A.Z-2 isoforms, their biochemical characterization quickly ensued [18,48]. In the absence of antibody-specific reagents for these isoforms, the initial search took particular advantage of the ability to distinguish them by SDS-PAGE in chicken (see Figure 1A) [41,48]. The gene for each of the isoforms was knocked out individually in the chicken DT40 B cell line. It was found that of the two complexes, SNF2-related CREBBP activator protein (SRCAP) and p400/Tip60, which in vertebrates are responsible for the deposition of H2A.Z, presumably through their common histone chaperone subunit transcription factor like 1 (YL1), SRCAP did not exhibit any specificity for either isoform, and in chicken this particular complex is responsible for about 70% of their deposition onto chromatin [49]. However, it was observed that H2A.Z-2 deficient cells proliferated 20–30% more slowly than wild type or H2A.Z-1 deficient cells [48]. The lack of isoform specificity for their deposition onto chromatin is not surprising as the sites of their interaction with YL1 are outside of the regions where their amino acid sequences vary (Figure 1C). This is also most likely true for their removal from the nucleosome by acidic leucine-rich nuclear phosphoprotein 32 family member E (ANP32E), which is a member of the p400/Tip60 complex but not SRCAP [50]. These data do not preclude the existence of other still unknown tissue-specific H2A/H2A.Z exchanging complexes as will be discussed later, and a generalized conclusion is not yet possible.

Crystallographic analysis of nucleosome core particles (NCPs) reconstituted with the H2A.Z-1 and H2A.Z-2 isoforms subsequently demonstrated the existence of a structural polymorphism in their L1 loop regions as a result of the sequence variation at position 38 (Figure 1C) [47]. The L1 loop falls within the histone fold, and as such structural differences might have mechanistic consequences [2]. Yet, the lack of differences in stability of the H2A.Z-1 and H2A.Z-2 reconstituted NCPs in comparison to significant differences in their exchange rate observed in situ by fluorescence recovery after photobleaching (FRAP) in living cells does nothing but to complicate the enigma further: If the two isoforms exchange with different rates which are not affected by the NCP stability, the change in chromatin binding dynamics would need to be affected by their differential interaction with histone chaperones [47]. As described in the previous paragraph, to date there is no evidence for this.

Regardless of the structural ambiguity, differential functional elucidations have steadily been coming forward in recent years, especially as they pertain to individual isoform dysregulation in different diseases.

Indeed, the two isoforms have been shown to have different roles in several types of cancer [51]. For instance H2A.Z-1 exhibits a substantial increase in prostate cancer cell lines in response to treatment with androgen [52]. Of interest, the promoter regions of the H2A.Z isoform genes are very different, with that of H2A.Z-1 having a few myelocytomatosis (MYC) binding elements [18]. Hence, the increase observed in response to androgen is most likely the result of MYC transcription factor binding to these sites in response to androgen. It was also shown that H2A.Z-1 increased in a LNCaP xenograft model of castration-resistant prostate cancer [52]. Histone H2A.Z-1 has also been shown to have a regulatory role in the epithelial-mesenchymal transition in liver cancer [53]. Regulation of c-MYC by NF-ĸB is known to be involved in the epithelial-mesenchymal transition of cancer [54]. Histone H2A.Z-2 has been found to be a driver of malignant melanoma and was required for cellular proliferation and a mediator of drug sensitivity in this cancer [55,56]. The involvement of H2A.Z-2 in this instance might reflect the important role this particular isoform has in DNA damage, as this and its repair are important components of this type of cancer, especially as it pertains to UV-induced DNA damage [23,57,58]. Indeed, histone H2A.Z-2 sumoylation has been shown to regulate the exchange of this variant at the sites of DNA damage [59].

Besides cancer, the two H2A.Z isoforms play differential, context-specific roles in neuronal activity-induced transcription of immediate early genes [60]. Also, H2A.Z-2 has been shown to be responsible for the craniofacial defects arising from floating-harbor developmental syndrome (FHS) and hippocampal levels of the same isoform are affected by fetal alcohol spectrum disorder (FASD) [61,62].

## 2. Materials and Methods

### 2.1. Structure Prediction

Histone fold domain (HFD) structures of histone H2A.Z.1 and H2A.Z.2 were predicted using the Phyre2 server [63].

### 2.2. Phylogenetic Inference

Phylogenetic trees were reconstructed using nucleotide and protein H2A.Z-1 and H2A.Z-2 sequences from the spider *Stegodyphus tentoriicola*, the mussel *M. galloprovincialis* and from human (H2A.Z.1 and H2A.Z.2.1). Sequences were aligned using the BioEdit program [64] and used in the reconstruction of neighbor-joining phylogenetic trees based on *p* distances using the program MEGA version 6 [65]. The reliability of the tree topology was contrasted using bootstrap analysis (1000 replicates), and human and mussel canonical H2A sequences were used as outgroups. In silico analysis of H2A.Z-1 and H2A.Z-2 intron regions were performed as in [42].

### 2.3. Nuclear Magnetic Resonance Spectroscopy

Histone peptides corresponding to residues 8–18 or 2–25 from human H2A.Z-1, human H2A.Z-2, and chicken H2A.Z-1 were synthesized by GL Biochem (Shanghai). Peptides were prepared at 2 mM in [^2^H] Tris, pH 7.5, with 150 mM NaCl and either 10% or 99% D_2_O.

NMR spectroscopy used an 800 MHz Bruker Avance spectrometer equipped with a triple-resonance gradient cryogenic probe, with data collected at 298 K. Data collection and processing used TopSpin 3.2 (Bruker) followed by additional processing with NMRPipe/draw [66]. ^1^H, ^13^C and ^15^N chemical shift assignment used the CCPN program [67] with figures produced in Sparky 3 (T.D. Goddard & D.G. Kneller, University of California, San Francisco, CA, USA). Chemical shift assignment used 2 mM peptide samples in NMR buffer as described above, with 2D ^1^H,^1^H-TOCSY spectra in 10% and 99% ^2^H_2_O (mixing time of 80 ms), accompanied by natural abundance 2D ^1^H,^15^N-HSQC spectra in 10% ^2^H_2_O and 2D ^1^H,^13^C-HSQC spectra in 99% ^2^H_2_O. Investigation of peptide conformation used 2D ^1^H, ^1^H-NOESY spectra in 10% ^2^H_2_O with a mixing time of 200 ms.

### 2.4. Tissues, Cell Lines, Preparation of Nuclei, and Chromatin Fractionation

Mouse (CD-1) brains and livers were collected at different developmental times ranging from E14 to P60 as described in [68]. Chicken red blood cells (RBC) were prepared as described elsewhere [69]. Marek’s virus-transformed MSB cells (Marek’s chicken spleen lymphoma induced by the BC-1 strain) were grown in 5% fetal calf, 5% newborn serum in 1:1 Dulbecco’s modified Eagle’s medium/RPMI 1640 medium supplemented with 50 mM HEPES, 30 mM bicarbonate, and 2 mM glutamine as described previously [70]. The cells were grown to a density of 1–2 × 10^6^ cells/mL and then were harvested or incubated in the presence of 5 mM sodium butyrate for 20–22 h before harvesting [70].

MSB and RBC nuclear preparations were carried out as described previously [69,70]. For chicken and mice brain and livers as well as for chicken testis, nuclei were prepared as follows: all steps were carried out on ice unless otherwise mentioned. Frozen tissue was homogenized in cell lysis buffer (4 volumes per gram of tissue), ‘Buffer A’, (0.25 M Sucrose, 60 mM KCl, 15 mM NaCl, 10 mM MES (pH 6.5), 5 mM MgCl_2_, 1 mM CaCl_2_, 0.5% triton X-100) containing protease inhibitor cocktail at a dilution of 1:100 (Roche Molecular Biochemicals, Laval, QC, Canada). After a 10 min incubation on ice, the sample was centrifuged at 600× *g* for 5 min at 4 °C. Pellet was re-suspended in 4 volumes of Buffer A per gram of tissue. The suspension was centrifuged at 600× *g* for 5 min at 4 °C. Pellet was re-suspended in 4 volumes per gram of tissue of the nuclei resuspension buffer, or ‘Buffer B’, (50 mM NaCl, 10 mM Pipes (pH 6.8), 5 mM MgCl_2_, 1 mM CaCl_2_) and centrifuged at 600× *g* for 5 min at 4 °C. The final nuclear pellet was then re-suspended in 2 volumes of Buffer B and the DNA concentration of the sample was determined by measuring the absorbance at 260 nm (OD260). To this end, 5 μL of the nuclear suspension were hypotonically lysed in 975 μL distilled water while vortexing followed by addition of sodium dodecyl sulphate (SDS) to a final concentration of 0.2% and the absorbance determined using an extinction coefficient A260 DNA = 20 cm^2^ mg^−1^ [71]. In the case of chicken testis the nuclei were next acid extracted by Dounce homogenization in 0.6 N HCl. The protein extract was then precipitated with 6 volumes of acetone and incubated overnight at −20 °C. The precipitate was recovered by centrifugation at 9300× *g* in an Eppendorf microcentrifuge. The pellet was resuspended (washed) with 3 volumes of acetone at room temperature and centrifuged again under the same conditions. The final pellet was vacuum speed dried and used for gel electrophoretic analysis.

Chromatin fractionation was carried out as described in [72]. Briefly, purified nuclei were digested with micrococcal nuclease and centrifuged at low speed to generate a soluble chromatin fraction (SI). The remaining pellet was then hypotonically lysed and centrifuged again to release a fraction (SE) and an insoluble pellet (P). Salt extraction of histones was carried out as follows: sodium butyrate treated (5 mM) and non-treated MSB nuclei aliquots were resuspended in 10 mM Tris-HCl (pH 7.5) 0.5 mM EDTA with different NaCl concentrations at a final chromatin concentration of approximately 2.5 mg/mL. The samples were then incubated on ice for 20 min and centrifuged in an Eppendorf microcentrifuge at 16,000× *g* for 10 min at 4 °C. The pellets obtained were directly used for SDS-PAGE analysis and the supernatants were extensively dialyzed against distilled water at 4 °C and lyophilized before SDS-PAGE analysis. The SI fraction consisting mainly of histone H1 depleted nucleosomes was used for the analysis of their salt (ionic strength)-dependent stability using sucrose gradients prepared as described in [73] and made in buffers at different NaCl concentrations.

### 2.5. Cell Cycle Dependence

MSB cells were synchronized with thymidine block at ‘0 h immediately after the thymidine block, and at 2 and 4 h after release of the thymidine block. These samples were compared to an unsynchronized sample for the western blot analyses. For western blot analysis, samples were harvested at 1.5 and 4.5 h after release from thymidine block.

### 2.6. Electrophoretic Analysis

Proteins were analyzed using different types of polyacrylamide gel electrophoresis (PAGE): SDS-PAGE (15% acrylamide, 0.4 bis-acrylamide) [74], Acetic acid Urea (AU)-PAGE (5% acetic acid, 15% acrylamide, 0.1% bis-acrylamide, 2.5 M urea) [75]. Separation of the chicken H2A.Z-1 and H2A.Z-2 was achieved using 16 cm long SDS-PAGE (0.75 mm thick) ran at 210 V for 8 h at room temperature.

### 2.7. Western Blotting

SDS-PAGE separated proteins were transferred to a nitrocellulose membrane (GE Healthcare Life Sciences). The membrane was probed with different antibodies at indicated dilutions: H2A.Z 1:2000 (Ab 1474, Abcam, Toronto, ON, Canada), and H4 1:50,000 (rabbit serum produced in-house), followed by incubation with a secondary antibody conjugated to fluorescent dye, IRDye 800 Anti-rabbit IgG 1:10,000 (Rockland Antibodies & Assays, Gilbertsville, PA, USA). Immunoblots were scanned using Li-Cor Odyssey (LI-COR Biosciences). Images were analyzed using Li-Cor Image Studio Version 5.2 software.

### 2.8. RNA Extraction and cDNA Preparation of Liver and Brain

RNA extraction of liver tissue was performed using TRIzol^®^ reagent (Invitrogen, Carlsbad, CA, USA), following the manufacturer’s instructions. RNA extraction of brain tissue was performed using the RNeasy Mini Kit (Qiagen, Germantown, MD, USA), following the instructions provided by the supplier. 2 ug of extracted RNA was reverse-transcribed using High-Capacity cDNA Reverse Transcription Kit (Applied Biosystems, Foster City, CA, USA), following the manufacturer’s instructions.

### 2.9. Quantitative Reverse Transcription Polymerase Chain Reaction (qRT-PCR)

qRT-PCR was carried out in triplicate using SYBR Select Master Mix (Applied Biosystems, Foster City, CA, USA) using a Stratagene MX3005P qPCR system (Santa Clara, CA, USA). The list of primer pairs used for qRT-PCR is shown in the Table 1 below. Thermocycling conditions for all primer sets were: 9 min 95 °C followed by 40 cycles of 15 s 95 °C, 30 s 60 °C, and 45 s 72 °C. Relative mRNA levels were calculated as described previously [76].

### 2.10. Graphic Representation and Statistical Analysis

Statistical analyses were performed using GraphPad Prism version 5.0. Data are represented as mean ± standard error of the mean (SEM).

## 3. Results

### 3.1. NMR Studies Reveal the Degree of N-terminal H2A.Z-1 and H2A.Z-2 Structure

As described in the previous section, of the three amino acid differences between the HA.Z.1 and H2A.Z.2 isoforms (Figure 1C), only position 38 has been extensively studied. We therefore decided to look at the structural consequences of sequence variation at the very N-terminal end around position 14 using nuclear magnetic resonance (NMR) (Figure 3A). In this way, we were able to determine that such variability imparts significant changes in the behavior of the protein sequence surrounding amino acid 14 (Figure 3B). In particular, the region comprising residues 8 to 18 has the sequence KAKAKA in human H2A.Z.2, KAKTKA in human H2A.Z.1 and KTKTKA in chicken H2A.Z.1 (Figure 3B). The alanine-rich sequence (hH2A.Z.2) displays multiple-scale dynamics and may self-associate to some degree. The substitution of one alanine for threonine in hH2A.Z.1 reduces the dynamics in this region. Furthermore, the presence of two threonines in chH2A.Z.1 corresponds to the largest reduction in dynamics and no evidence of self-association (Figure 3B). Hence, n-terminal isoform variation of H2A.Z has very important implications for the flexibility and structure dynamics of this domain.

### 3.2. DNA Methylation, H2A.Z Exclusion, and Developmental Expression Changes of H2A.Z-1 and H2A.Z-2 in Mouse Liver and Brain

There is a well-established correlation between DNA methylation, a chemical mark associated with low gene expression, and the absence of histone H2A.Z (Figure 4A) which primarily results from the exclusion of H2A.Z from methylated DNA within gene bodies [31,78,79].

One of the DNA methylation readers, MeCP2 (methyl CpG binding protein 2) which can also bind mCH (where H = A or T or C), is highly abundant in the mature brain at approximately one molecule for every three nucleosomes and every two nucleosomes in neurons [73,80]. Neuron-specific long gene bodies are enriched in CA dinucleotides which, when methylated, are bound by MeCP2 (i.e.,: *Bdnf* (Brain Derived Neurotrophic Factor), see Figure 4A)), repressing their activity [81,82]. During brain development, levels of methylation in mouse increase from 2.9% (mainly CG) at birth to 4.2–4.5% in adulthood as MeCP2 steadily increases by approximately 50% (from E14 to P30) during neuronal differentiation [68,83]. A 1.3–1.6% increase in CH methylation corresponds to approximately three CH methylated sites per NCP on average. Thus, we reasoned that even if CA methylation amounts to one-third of this (which indeed is a lower estimate, as CAC is the preferential site of methylation in neurons [84]) there should still be plenty of internal methylation in this tissue capable of displacing H2A.Z during differentiation. Moreover, we decided to analyze the expression of the two different isoforms in brain and compare it to that observed in liver, a tissue containing only about 3–5% of the MeCP2 which is present in brain [73]. The results of these analyses are shown in Figure 4B.

In agreement with our prediction, the presence of H2A.Z exhibited a limited but noticeable decline of about 15–20% during brain development (Figure 4B_1_), whereas in liver it increased by almost double (Figure 4C_1_). Such a decrease of H2A.Z during brain development is in disagreement with earlier results from the Suau lab which state that the variants stay constant during rat brain development [85]. However, the experimental approach taken at the time (gel staining) might not have been sensitive enough to detect a 15% change. The amount of H2A.Z in the embryo stages of early development (E18) was about double in brain compared to liver and it evens up in late development (P30–P60) (results not shown). Interestingly, the expression of the gene for H2A.Z-1 decreased during development (Figure 4B,C_2_). This is in contrast to *H2a.z-2* gene expression, which increased during embryonic development, but decreased after birth (Figure 4B,C). The patterns exhibited by the two isoforms were very similar in both tissues (Figure 4B,C_2_,C_3_). The peak of expression of the H2A.Z-2 gene at birth (P0) (Figure 4B,C_3_, arrows) in contrast to the constant protein levels (Figure 4B,C_1_) may reflect a high protein turnover of this isoform at this point in development. Nevertheless, such sudden increase remains quite intriguing and, once again, it underscores the different functional attributes of the two isoforms.

### 3.3. Changes in H2A.Z-1/H2A.Z-2 Composition in Different Cell Types and during the Cell Cycle

As mentioned at the beginning of this chapter, chicken liver provided the source for the first identification of the H2A.Z isoforms (Figure 1). Hence, we took advantage of this biological system to analyze the distribution of the isoforms in different tissues of this organism (Figure 5A) and in different chromatin fractions (Figure 5B) as well as during the cell cycle using a chicken lymphoblastoid cell line (Figure 5C).

As seen in Figure 5A, chicken liver contains approximately equal amounts of H2A.Z-1 and H2A.Z-2 (see also Figure 6 L2/L3) whereas in brain and MSB (Marek’s chicken spleen lymphoma induced by the BC-1 strain) cells the amount of H2A.Z-1 is slightly higher. Of note, treatment of MSB cells with sodium butyrate to inhibit histone deacetylases (HDACs) produces an electrophoretic band of reduced mobility compared to H2A.Z-1 (see Figure 5A, red arrow). Chicken erythrocytes by contrast show only one predominant band (Figure 5A,B, red blood cells (RBC), Figure 6C, and chicken marker (CM)) with slightly less electrophoretic mobility than H2A.Z-2, and which might correspond to a posttranslational modification (PTM) of this isoform. The prevalent presence of this band in this chicken blood tissue deserves further analysis.

We next wanted to check the distribution of the two H2A.Z isoforms amongst chromatin fractions using a rudimentary method of euchromatin/heterochromatin partitioning (Figure 5B) [72]. Figure 5B shows that no major difference can be observed in the isoform distribution as a result of the chromatin fractionation regardless of the nature of the tissue or cell type used.

In contrast to the results above, the data from the previous section suggest that the H2A.Z isoforms function as a differentiation device, and we next wanted to explore whether they had any participation in the cell cycle. The results shown in Figure 5C are in agreement with the results of Matsuda et al. which observed no difference in the distribution of cells in the G1, S and G2/M phases amongst the H2A.Z-1/H2A.Z-2 deficient and wild type DT40 cells [48].

### 3.4. A Specific Role for One of the H2A.Z Isoforms in Spermatogenesis

The occurrence of two histone H2A.Z isoforms appears to have taken place several times in the course of animal evolution (Figure 2), and the amino acid variation seems to have occurred to different extents at around the same protein locations. However, the amino acid substitutions do not always coincide (Figure 2A). Therefore, the classification nomenclature (isoform 1 and 2) (Figure 3A) is rather arbitrary in structural terms and not necessarily equivalent. A good example is provided by mussel (*Mytilus* sp.) within the bivalve mollusks. It was shown in this group of animals that one of the two variants was highly specifically found in the male gonad [42]. This interesting observation prompted us to analyze the histone H2A.Z isoform distribution in the chicken male gonad (Figure 6).

Figure 6A shows an SDS-PAGE analysis of the testis histones in comparison to liver histones and Figure 6B shows the same analysis by Acetic acid urea (AU)-PAGE in order to identify the presence of chicken protamine in the testes extracts. Figure 6C is a histone H2A.Z western blot analysis of the SDS-PAGE shown in section A and Figure 6D shows a quantitative analysis of the data. The amount of H2A.Z-1 in this tissue is about 3-4 fold that of H2A.Z-2 which compares well with the 5-fold increase observed for the testis specific H2A.Z in the mussel.

### 3.5. Histone Isoforms H2A.Z-1 and H2A.Z-2 Confer Identical Ionic Strength-Dependent Chromatin Affinity and Nucleosome Stability In Situ

In the early analysis of potential structural differences imparted by the two histone H2A.Z isoforms to nucleosomes, the salt-dependent stability of NCPs reconstituted with recombinant histones including the two isoforms and 146 bp DNA was shown to be indistinguishable [47]. However, the possibility exists that in the native setting of the cell, a preferential association of the two isoforms with different factors (i.e., association with other histones carrying specific PTMs [18]) may substantially alter their individual association to chromatin and to the nucleosome. To address this possibility, the H2A.Z-1 and H2A.Z-2 composition of the supernatants (protein released from chromatin) and pellets (protein bound to chromatin) of nuclei from MSB cells extracted with different salt (ionic) concentrations, in the presence (+) or the absence (−) of sodium butyrate, was analyzed (Figure 7A). We also analyzed the isoform composition of native nucleosomes obtained from the SI chromatin fraction from MSB cells run in sucrose gradients under different salt concentrations (Figure 7B). As both figures indicate, we were unable to detect any difference in either one of these approaches.

## 4. Discussion

Genetic complementation analysis suggests that the N-terminal end of H2A.Z, and hence the amino acid sequence variation of its two main isoforms (H2A.Z-1 and H2A.Z-2), can distinctively contribute to the multiplicity of its epigenetic regulations [87]. The N-terminal domain of RD H2A binds to nucleosome DNA at two locations, centered around approximately 40 bp from the nucleosome dyad [39]. Although RD H2A and RI H2A.Z primary sequences exhibit some differences within this region, the sequence motif _11_KAKT_14_ in H2A.Z is highly conserved in H2A and it interestingly includes the amino acid at position 14 which is where the variation between the H2A.Z-1 and H2A.Z-2 isoforms occurs (see Figure 2A and Figure 3A). Because H2A.Z and H2A occupy the same location in the nucleosome, it is likely that their N-terminal interactions with DNA are also very similar [37,38]. As our NMR studies indicate (Figure 3), the amino acid sequence variation between the H2A.Z-1 and H2A.Z-2 isoforms, within this domain, imparts it with a distinctive conformation (Figure 3B) which can be anticipated from the different altered electrophoretic mobilities exhibited by these two proteins in chicken (Figure 1A) and [48]. The different flexibility of this region between the two isoforms can be further modulated by PTMs. Indeed, it can become heavily acetylated [72]. Therefore, all of this might have an important impact in the way the N termini of the two H2A.Z isoforms interact with DNA in the nucleosome. This does not necessarily need to invoke a change in NCP structure or stability by itself, but might play a role in mediating the action of chromatin remodeling complexes such as switching of the mating type/sucrose non fermenting (hSWI/SNF) [88].

An important observation described above is the prevalent presence of one of the isoforms during spermatogenesis (Figure 6) which, in vertebrates, corresponds to H2A.Z-1 (Figure 6C,D). Although the results shown here do not identify the stages of this differentiation and accumulation process, it is interesting to note that H2A.Z-1 has been shown to be present in human spermatozoa [89]. It is tempting to speculate that the major role of H2A.Z, and in particular the enrichment in H2A.Z-1, may involve the events taking place during meiosis. For example, in yeast (*S. cerevisiae*), H2A.Z participates in sister chromatid cohesion, and, in fission yeast, H2A.Z promotes DSB formation and initiation of meiotic recombination [90,91]. Indeed, this would agree with the binding preference of the vertebrate H2A.Z-1 isoform to Brd2, as Brd2 has been shown to be expressed in diplotene spermatocytes [13,92]. Moreover, Brd2 is a transcriptional co-activator/co-repressor associated with SWI/SNF [93].

For most of the other results presented here, we do not see a large difference neither in the structural nor in the functional genomic distribution of the two isoforms. Structurally, we do not see any significant variation in the binding affinity between H2A.Z-1 and H2A.Z-2, neither at the nucleosome level, nor at the chromatin fiber, even when the global extent of histone acetylation is increased by treatment with the HDAC inhibitor sodium butyrate (Figure 7). Therefore, any changes in the DNA binding affinity between the two isoforms must reside on molecular mechanisms other than those specified by their amino acid differences. Several alternative scenarios can be envisaged. Firstly, differential PTMs on each isoform in the native setting and/or their association with other histone PTMs; indeed, acetylation of H2A.Z significantly affects the binding affinity of H2A.Z to chromatin [94]. Hence, differential acetylation of the two isoforms could alter their different functional outcome. For instance, the flexibility of the n-terminal tail between different isoforms can be enhanced by their specific acetylation, regardless of the lack of difference observed in the acetylation rate of H2A.Z-1 and H2A.Z-2 by p300 [95]. Also, it has been shown that H2A.Z-2 preferentially associates with H3K4me3 [18]. Similarly, their different association with other histone variants (i.e., H3.3) might play a role [96,97]. Finally, their interaction with specific partners such as Bdr2 or others such as those recently identified could impart differential function [13,98].

At the functional level, the two isoforms exhibit a similar level of expression in brain and in liver (Figure 4), and no difference could be detected at the different phases of the cell cycle (Figure 5C), in agreement with the early results of Matsuda et al. and their isoform knock down experiments [48]. Also, we did not see any obvious euchromatin (SI) versus heterochromatin (SE/P) chromatin partitioning (Figure 5B). The latter could partly be because, as the chromatin immunoprecipitation-sequencing (ChIP-seq) experiments performed in [61] seem to indicate, such a difference is much less qualitative than quantitative. This explanation however, is quite surprising, especially since any direct role of H2A.Z and, hence its isoforms, in transcription activity is still extremely debatable. In this regard, a good closing example of this is provided by two very recent seemingly contradictory reports which have come up at the time of writing this review. In the first, it is indicated that in yeast, H2A.Z is almost exclusively incorporated into the +1 nucleosome in the direction of transcription, suggesting that H2A.Z has a causative role in the transcription initiation process at this location [99]. In the second, using H2A.Z KO post-mitotic muscle cells, it is shown that this histone variant is neither required to maintain nor to activate transcription which would be in line with the concept that H2A.Z is not a direct player but rather a marker of transcription [11,100].

## 5. Concluding Remarks

It is quite apparent from what has been described above that the functional relevance of the two H2A.Z-1 and H2A.Z-2 isoforms is complex, and that a lot of open questions remain. How the structural change imparted by sequence variation at position 14 (Figure 3) and that observed for the alterations in the L1 loop of the histone fold (HFD) at position 18 (see Figure 1C) account for different functionality of the isoforms is not yet clear as the effects of such changes in the structural subunit of chromatin, the nucleosome, and chromatin itself are not obvious (Figure 7) [2,47]. It is equally not obvious how such changes might affect the protein interacting partners of these isoforms.

Although the expression of the two isoforms appears to be selectively misregulated in several cancers (de-differentiation processes) and other diseases, their differential functional components under normal conditions are not as clear cut, and they often appear to be redundant [48,61,101,102]. In particular, the lack of a significantly different chromatin functional partitioning during the cell cycle (Figure 5C) is in agreement with their rather uniform genome wide distribution (Figure 5B), which may be preferentially altered depending on the differentiation cues of the tissue [48]. Consistently, the most striking differences we observed in isoform composition are those arising from developmental processes such as chicken erythrocyte differentiation or spermatogenesis (Figure 6), and it thus appears that one of the major roles of H2A.Z-1 and H2A.Z-2 might be preferentially geared to different aspects of cell differentiation and development [100,103].

## Figures and Tables

**Figure 1 cells-09-01167-f001:**
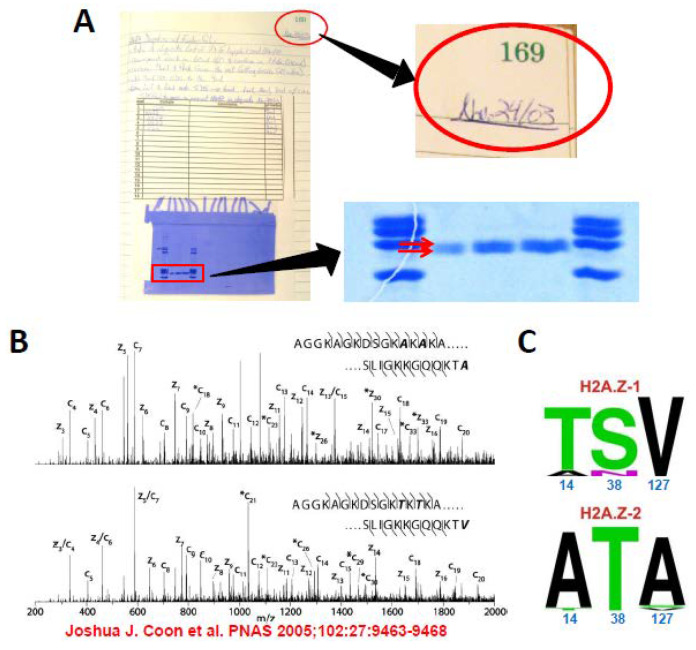
(**A**) First evidence (2003) for purified chicken liver histone H2A.Z running as a double band in SDS-PAGE. (**B**) Identification of the two bands corresponding to H2A.Z-1 and H2A.Z-2 using sequential ion/ion reactions and tandem mass spectrometry [41]. (**C**) Logos representing the mammalian amino acid residues at positions 14, 38, and 127 in H2A.Z-1 and H2A.Z-2 [18].

**Figure 2 cells-09-01167-f002:**
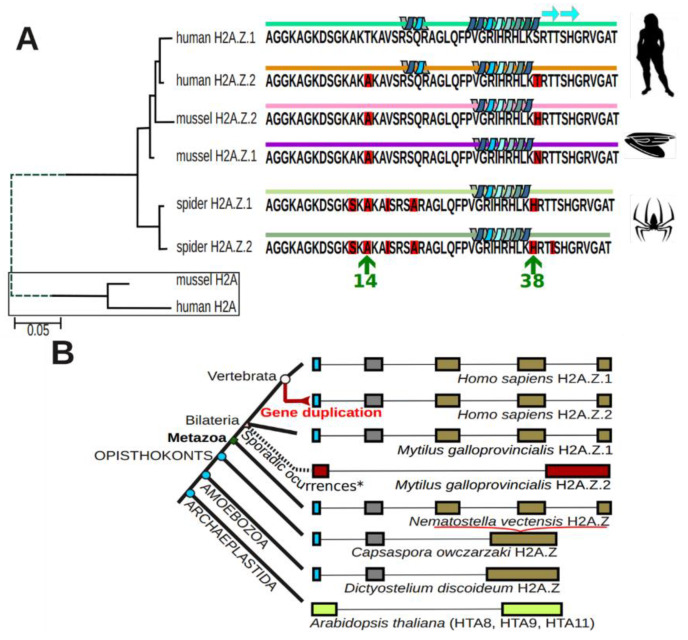
(**A**) Phylogenetic relationship among histone H2A.Z isoforms in Metazoa. A.Z-1 and H2A.Z-2 isoforms have a closer intraspecific than interspecific evolutionary relationship. (**B**) Major evolutionary transitions in histone H2A.Z organization in eukaryotes. Exon homology was inferred from the sequence length among different groups and represented by identical color. An increasing exon complexity appeared in the transition from protists to metazoans where possibly exons 3, 4, and 5 arose from exon 3 in protists (*Capsapora owczarzaki* and *Dictyostelium discoideum*). Introns are represented with arbitrary scales for comparative purposes. Figure adapted from Figures 4 and 5 in [42].

**Figure 3 cells-09-01167-f003:**
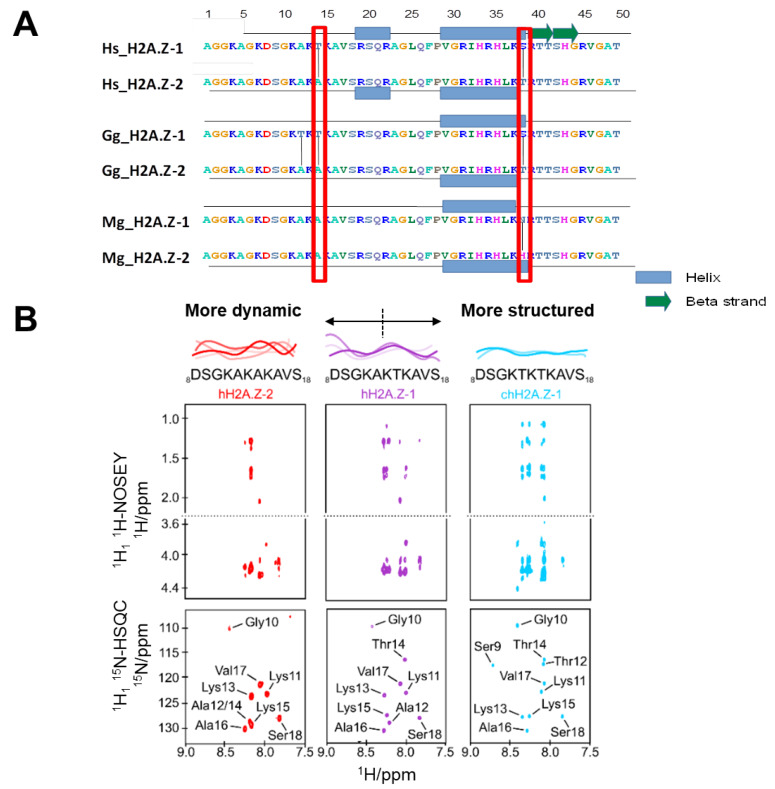
(**A**) Amino acid sequence of the first 49 amino acids of H2A.Z-1 and H2A.Z-2 isoforms and schematic representation of the secondary structure prediction. The red boxes highlight the sites of amino acid sequence variation and the red arrow points to the extra site of variation in chicken H2A.Z-1. Hs: *Homo sapiens*; Gg: *Gallus gallus* (chicken); and Mg: *Mytilus galloprovincialis* (blue mussel). (**B**) Variable dynamic properties within the H2A.Z N-terminal tails. Peptides corresponding to residues 8–18 in human H2A.Z.2, human H2A.Z.1 and chicken H2A.Z.1 were characterized by NMR spectroscopy. The hH2A.Z-2 peptide has all alanines in the KAKAKA region, and by NMR spectroscopy displays only a limited number of nuclear Overhauser effect (NOE) cross-peaks, as well as significant broadening of the amide cross-peaks in the ^1^H, ^15^N-HSQC spectrum. These findings are consistent with a range of conformational dynamics and may also suggest a degree of self-association. In contrast, the chH2A.Z.1 sequence with two threonines (KTKTKA) displays a clear reduction in side chain dynamics, as evident by an increased number of NOE cross-peaks. The uniform and well-defined backbone amide cross-peaks in the ^1^H, ^15^N-HSQC (heteronuclear single quantum coherence) spectrum also suggest reduced conformational flexibility and no evidence of self-association. The hH2A.Z.1 sequence with a single threonine (KAKTKA) displays intermediate properties. The ^1^H, ^15^N-HSQC cross-peaks have been annotated by residue type and number.

**Figure 4 cells-09-01167-f004:**
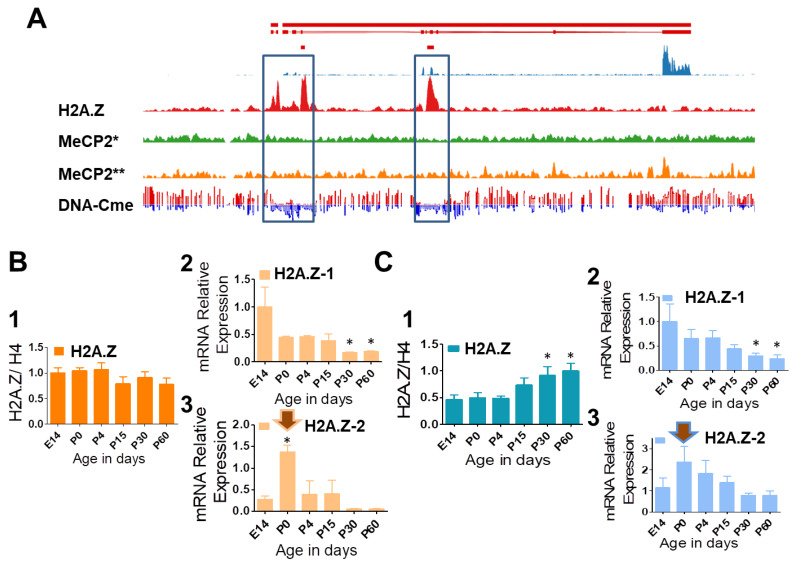
(**A**) Schematic view of the genomic (solid red line) and coding (thin red line) representation of the BDNF gene and integrative genomics viewer (IGV) profiles of: mRNA–seq (blue) [86]; H2A.Z (red) [86]; MeCP2* (green) [82]; MeCP2** (orange) [80]; whole-genome bisulfite sequencing (red and blue): red reads are 5mC and blue reads are unmethylated cytosines. Notice that H2A.Z binds to unmethylated C (dataset provided by Andrew J. Kennedy and Kristine E. Zengeler). * https://www.ncbi.nlm.nih.gov/geo/query/acc.cgi?acc=GSE60071. ** https://www.ncbi.nlm.nih.gov/geo/query/acc.cgi?acc=GSM494290. (**B**,**C**) Developmental H2A.Z protein expression via quantitative western blot (1) and isoforms H2A.Z-1 and H2A.Z-2 relative mRNA expression levels via qRT-PCR (2, 3) in mouse brain (B) and liver (C). One-way ANOVA with Dunnett’s multiple comparison was performed to analyze western blot (*n* = 3) and qRT-PCR data (*n* = 5). E14 was used as a reference. Significance: * *p* < 0.05.

**Figure 5 cells-09-01167-f005:**
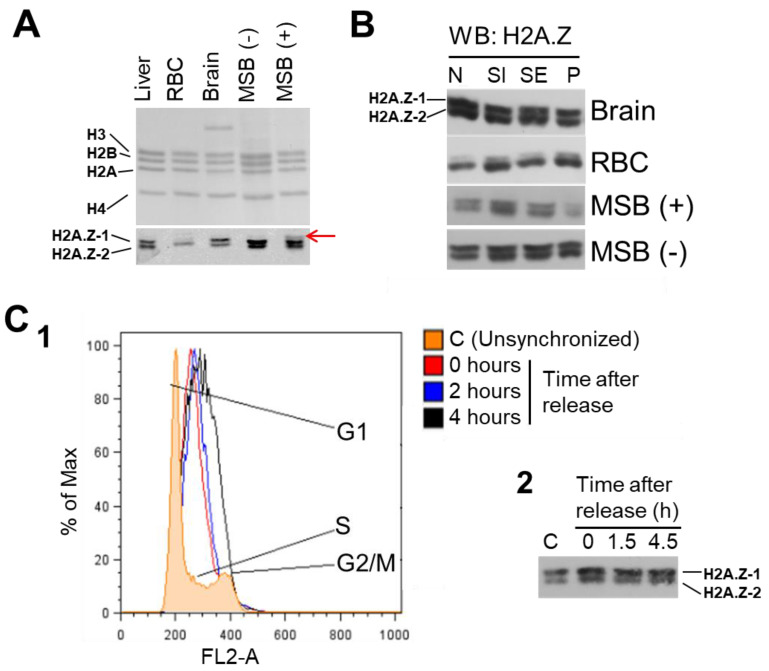
(**A**) Chicken tissue, (**B**) chromatin, and (**C**) cell cycle distribution of chicken H2A.Z-1 and H2A.Z-2 isoforms in MSB cells before (-) and after (+) sodium butyrate treatment. The red arrow in (A), points to a band with higher mobility than H2A.Z-1 which appears after sodium butyrate treatment. (B) N = starting nuclei; SI = soluble chromatin fraction obtained immediately after micrococcal nuclease (MNase) digestion; SE = hypotonically released chromatin fraction; P = insoluble chromatin pellet; MSB = Marek’s transformed chicken cells; RBC = red blood cells (chicken). (C_1_) Distribution of cell cycle phases in unsynchronized (orange) and synchronized MSB cells after 0 (red), 2 (blue), and 4 (black) hours of release from thymidine block. Cells display typical proportional increase in the S phase after synchronization and progression to G2/M phase. (C_2_) Western blot of the H2A.Z composition in MSB cells at different times after release from thymidine block.

**Figure 6 cells-09-01167-f006:**
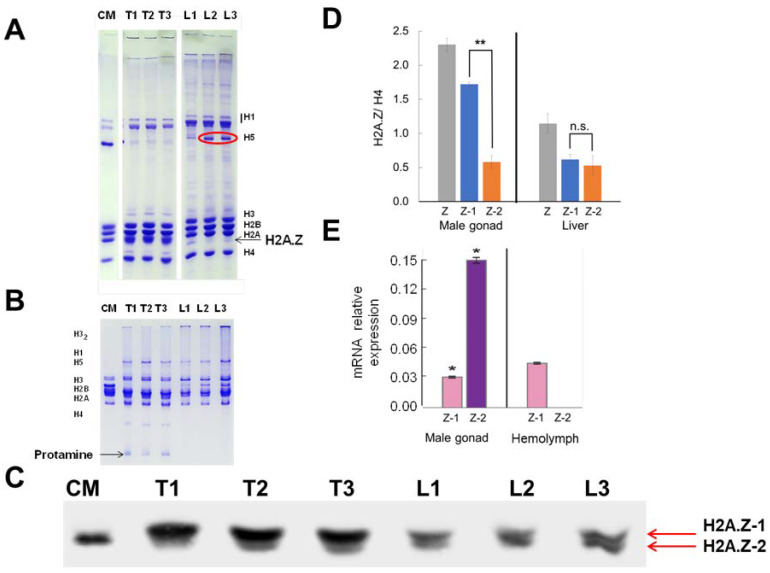
One of the main H2A.Z isoforms is preferentially expressed during spermiogenesis. (**A**) Long SDS-PAGE of three chicken testes and livers. The red ellipsoid highlights the presence of some histone H5 in liver, resulting from different extent of blood contamination. (**B**) AU-PAGE analysis of the same samples in (A) showing the presence of chicken protamine in the testes samples. (**C**) Western blot analysis of (A). (**D**) Bar plot representation of the results shown in (C). (**E**) Bar plot of the relative mRNA expression of the H2A.Z-1 and H2A.Z-2 isoforms in the mussel *Mytilus* sp. (adapted from [42]). Data are represented as Mean ± SEM. Unpaired two-tailed *t*-tests determined significance. * *p* < 0.05, ** *p* < 0.01.

**Figure 7 cells-09-01167-f007:**
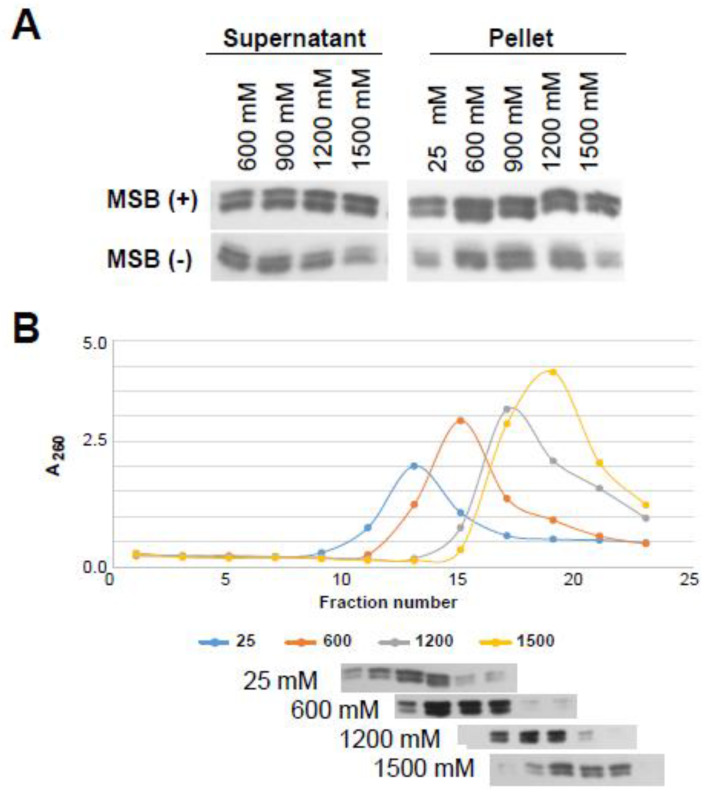
Ionic (NaCl) strength-dependent affinity of H2A.Z-1 and H2A.Z-2 chicken isoforms in (**A**) chromatin and (**B**) nucleosomes.

**Table 1 cells-09-01167-t001:** Mouse RT-qPCR primer sequences used in the study.

Gene Name	Forward Primer Sequence	Reverse Primer Sequence	Annealing Temperature (°C)	References
H2A.Z-1	CACCGCAGAGGTACTTGAGTT	TCCTTTCTTCCCGATCAGCG	60	
H2A.Z-2	CAAGGCTAAGGCGGTGTCTC	CTGCTAACTCCAACACCTCAGC	60	Matsuda et al., 2010 [48]
GAPDH	AACGACCCCTTCATTGAC	TCCACGACATACTGAGCAC	60	
TBP	CCCCACAACTCTTCCATTCT	GCAGGAGTGATAGGGGTCAT	60	Martinez de Paz et al., 2015 [77]

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
