# Peer review of "Deciphering the Enigma of the Histone H2A.Z-1/H2A.Z-2 Isoforms: Novel Insights and Remaining Questions"

_cells, 2020, doi:10.3390/cells9051167_

Round 1

Reviewer 1 Report

In this manuscript the authors present a combination of a literature review and experimental data to create an up-to-date work on the H2A.Z-1 and H2A.Z-2 histone variants.

The introductory review is a detailed compilation of the currently known data about H2A.Z variants from yeast to human. The experimental section is mainly a biochemical approach of the two variants (with a very well described materials and methods section) to have a broad view of the possible structural and functional differences of the two variants.

Only minor comments that authors can address:

1) In the evolutionary analysis, Arabidopsis H2A.Z is included, but nothing is explained in the text. I understand that the review is focused in the animal H2A.Z variants, but maybe functions and singularities of H2A.Z in an outsider group as plants can give light to some of the evolutionary questions. Besides that, authors can consider attracting a wider number of readers and citations if H2A.Z in plants is considered.

2) In some figures is difficult to read what is written. For example, the axis in Fig. 4B, the species and sequence in the Fig. 2 and Fig. 3A, or the panels in Fig. 3B.

3) Statistical tests in the expression analysis of Fig. 4 and Fig. 6D are missing.

4) Line 33. The two variants differ in 3-5 aa. If I understood correctly the color code in Fig. 2A (red marked aa), in spider there are 6 aa differences.

5) Line 57. When is said H2A.Z represents a 5% of the total RD H2A histones, it is unclear what this means, as H2A.Z is RI. Please, rephrase.

6) Line 294. The reader is addressed to the Fig. 4A to see the binding of MeCP2 to long genes, but this enrichment is not clear in the figure.

7) Line 302. “at issue” should be “a tissue”.

8) Line 313. Why in this sentence H2A.Z-2 is referred as “H2a.z-2” in italics?

9) Line 336 and 441. Why in these sentences H2A.Z-2 is referred underlined?

10) Line 340. “fig.” should be “Fig.”

11) Line 353. It is not clear to me the lines in the panel 5C-1 nor the table. It is not detailed in the legend.

12) What the asterisks in Fig. 6E means is not detailed in the legend.

Author Response

 1) In the evolutionary analysis, Arabidopsis H2A.Z is included, but nothing is explained in the text. I understand that the review is focused in the animal H2A.Z variants, but maybe functions and singularities of H2A.Z in an outsider group as plants can give light to some of the evolutionary questions. Besides that, authors can consider attracting a wider number of readers and citations if H2A.Z in plants is considered.

This is a good suggestion and it has been addressed on lines 101-120 and 148-161 of the revised version of the manuscript.

2) In some figures is difficult to read what is written. For example, the axis in Fig. 4B, the species and sequence in the Fig. 2 and Fig. 3A, or the panels in Fig. 3B.

These figures have been modified accordingly. The lettering in these figures has been adjusted to a larger, more readable font.

3) Statistical tests in the expression analysis of Fig. 4 and Fig. 6D are missing.

These have been included.

4) Line 33. The two variants differ in 3-5 aa. If I understood correctly the color code in Fig. 2A (red marked aa), in spider there are 6 aa differences.

This has been addressed (see line 33).

5) Line 57. When is said H2A.Z represents a 5% of the total RD H2A histones, it is unclear what this means, as H2A.Z is RI. Please, rephrase.

The statement has been clarified (lines 72-73).

6) Line 294. The reader is addressed to the Fig. 4A to see the binding of MeCP2 to long genes, but this enrichment is not clear in the figure.

The misunderstanding was created by an improper writing of the sentence which meant to say highly enriched in the CA dinucleotide. This has also been corrected (lines 378-380).

7) Line 302. “at issue” should be “a tissue”. 8) Line 313. Why in this sentence H2A.Z-2 is referred as “H2a.z-2” in italics? 9) Line 336 and 441. Why in these sentences H2A.Z-2 is referred underlined? 10) Line 340. “fig.” should be “Fig.”

The typographical errors have been corrected.

11) Line 353. It is not clear to me the lines in the panel 5C-1 nor the table. It is not detailed in the legend.

The figure has been reformatted to be more intuitive to the reader and a more detailed explanation is provided in the figure legend.

12) What the asterisks in Fig. 6E means is not detailed in the legend.

This has been addressed.

Reviewer 2 Report

Review of Cells-755125 Deciphering the enigma of the histone H2A.Z 1/H2A.Z-2 isoforms. Novel insights and remaining questions
Authors Cheema et al
Reviewer Herve Seligmann

In principle, this ms could be published after the points listed below are delat with. As it stands, it is not publishable. The mess regarding figure placement and numbering makes evaluating the results difficult.

1. general background on histones , structure, conservation etc, and functions,totally missing. An overview of histones in general is essential. Then state differences with/particularities of the focal histones analysed here.
2. general scheme describing histone structure, in complex with dna, and interactions with molecules inducing/regulating exrpession, and with chaperones, would help.
3. same as above, but specifically for H2A.Z isoforms
4. end of introduction: describe what the experiments will test for, and why

In the discussion, I find figure 2, which is after fiigures 4 and 5. I did not yet see figures 1 and 3. Figure 2 is too small, its siye must be increased for visibility
Figure 3 is after figire 2, but also after figures 4 and 5. Figure 1 still not found.
Figures 6 and 7 mentioned in the conclusion, and in opposite order, 7 before 6. both points are inadequate.

Author Response

1. general background on histones , structure, conservation etc, and functions,totally missing. An overview of histones in general is essential. Then state differences with/particularities of the focal histones analysed here.

This has been included at the beginning of the “Preamble” section.

2. general scheme describing histone structure, in complex with dna, and interactions with molecules inducing/regulating exrpession, and with chaperones, would help. 3. same as above, but specifically for H2A.Z isoforms. 4. end of introduction: describe what the experiments will test for, and why

We feel that this does not belong to the current manuscript and wherever this information was required in the text we have already included it there. The reviewer is referred to another review on the histone variant topic published by our own group: Cheema MS, Ausió J. (2015) The Structural Determinants behind the Epigenetic Role of Histone Variants. Genes (Basel).;6(3):685-713.

In the discussion, I find figure 2, which is after fiigures 4 and 5. I did not yet see figures 1 and 3. Figure 2 is too small, its siye must be increased for visibility Figure 3 is after figire 2, but also after figures 4 and 5. Figure 1 still not found. Figures 6 and 7 mentioned in the conclusion, and in opposite order, 7 before 6. both points are inadequate.

We have already referred to the problem with the scrambled figures in the publisher’s editing of our manuscript. This has now been addressed in the revised version.

Reviewer 3 Report

The manuscript by Cheema MS. and colleagues reports novel insights regarding histone variants H2AZ-1 and H2AZ-2. It also provides a load of information on our current understanding of the function(s) of these two variants in the regulation of transcription and chromatin organization. In sum, the manuscript is a mix between a review and a research article, as indicated by the authors at the beginning of the document.

I am not convinced by such format. I am even less convinced by the architecture of the document and its interest to the community. It is unclear to me whether the manuscript reports new data produced by the labs or whether it is a re-analysis of previously produced data. For instance, several information, presented as new, are actually reproducing previously published studies. This is clearly the case for Figures 4C, 5C, 2, 6E. I would thus recommend to re-format the document as a review and includes these « new » information as an illustration to highlight new ideas and hypothesis; or unveiling new concepts implicating H2AZ variants.

Major points:

On the biochemical part several controls are lacking:

  • 5B: authors must provide controls/markers showing the proper purification and fractionation of N, SI, SE and P.
  • 5C1: the figure is confusing. Please provide individual panels for each experimental condition as well as a quantification of 3 independent biological replicates.
  • 5C2: please provide cell cycle and DNA damage markers levels to confirm cell cycle re-entry and the cell cycle phase reached by the cells at each time point. Is H2AZ1/2 constant throughout the cycle? during the G1/S transition? in DNA damaged cells?

Minor comments:

-Not sure I understand lane 288-290. Is H2AZ or DNA methylation occurrence correlated with low gene expression?

-Poor quality documents / images are too small.

-Why are figure 2 and 3 located in the discussion section?

Author Response

I am not convinced by such format. I am even less convinced by the architecture of the document and its interest to the community. It is unclear to me whether the manuscript reports new data produced by the labs or whether it is a re-analysis of previously produced data. For instance, several information, presented as new, are actually reproducing previously published studies. This is clearly the case for Figures 4C, 5C, 2, 6E. I would thus recommend to re-format the document as a review and includes these « new » information as an illustration to highlight new ideas and hypothesis; or unveiling new concepts implicating H2AZ variants.

Figures 4C, 5C have never been published earlier.  Figures 2 and 6E have been modified from previous publications from our own lab and this is ok in a review. Moreover, where figures have been adapted, this is clearly indicated in the corresponding figure legend. We acknowledge that this reviewer does not like the organization of our manuscript which combines an introductory review followed by several of our unpublished data in the topic. This is unfortunate but represents this reviewer’s subjective opinion which obviously represents neither that of the two previous reviewers nor those of the eighteen authors of the paper.  

Major points:

On the biochemical part several controls are lacking:

  • 5B: authors must provide controls/markers showing the proper purification and fractionation of N, SI, SE and P.

Our lab has published numerous research papers using this fractionation and hence we do not feel this needs further information. The following papers are just a few examples: Dryhurst D, Ishibashi T, Rose KL, Eirín-López JM, McDonald D, Silva-Moreno B, Veldhoen N, Helbing CC, Hendzel MJ, Shabanowitz J, Hunt DF, Ausió J. Characterization of the histone H2A.Z-1 and H2A.Z-2 isoforms in vertebrates.  (2009 BMC Biol. 7:86. doi: 10.1186/1741-7007-7-86/ /// Thambirajah AA, Dryhurst D, Ishibashi T, Li A, Maffey AH, Ausió J. (2006) H2A.Z stabilizes chromatin in a way that is dependent on core histone acetylation. J Biol Chem. 281(29):20036-44

  • 5C1: the figure is confusing. Please provide individual panels for each experimental condition as well as a quantification of 3 independent biological replicates. 5C2: please provide cell cycle and DNA damage markers levels to confirm cell cycle re-entry and the cell cycle phase reached by the cells at each time point. Is H2AZ1/2 constant throughout the cycle? during the G1/S transition? in DNA damaged cells?

This is the way these data are usually represented in conventional research papers. The results were reproduced several times although the precise times of data collection were slightly different. The reviewer is referred to a research paper on H2A.Z-1 and H2A.Z-2 using this technique:  Matsuda R, Hori T, Kitamura H, Takeuchi K, Fukagawa T, Harata M.(2010) ) Identification and characterization of the two isoforms of the vertebrate H2A.Z histone variant. Nucleic Acids Res. 2010 Jul;38(13):4263-73. Moreover, as we indicate in the text, our results are of a confirmatory nature as they fully agree with those of the Harata lab. Hence, we believe there is no need to dig into these data any further.

Minor comments:

-Not sure I understand lane 288-290. Is H2AZ or DNA methylation occurrence correlated with low gene expression?

This has been addressed (lines 372-374).

-Poor quality documents / images are too small.

This has been addressed.

-Why are figure 2 and 3 located in the discussion section?

This has to do with the figure scrambling to which we have been referring.

Round 2

Reviewer 2 Report

-

Author Response

Dear reviewer,

Upon the advice of the editor, we have addressed the minor concern about the “general scheme describing histone structure in complex with DNA” requested by reviewer number 2 (see new lines 125-135). Notice that because histones do not interact independently with DNA but only in the context of the nucleosome, we are providing information within this frame.

Sincerely,

Juan Ausió, on behalf of all the authors

Reviewer 3 Report

I was really enthousiastic when I received the invitation to review this manuscript. I read it with a lot of interest and learnt a lot regarding the functions of H2AZ variants. Still, I was not convinced by the format and I thus suggested an editing of the review to better incorporate the complementary data in the review (such as in other journals such as Trends Journals or BioEssays). I fully agree this is my own opinion.

That said, authors claim that figures have never been published before. However, figures are reporting reproduction of previously published work (which is good) but without adequate controls, arguing that they are not necessary as similar results are already published (which is questionable).

I thus clearly not understand why these figures are presented in a format of a research article (suggesting these are new experiments) rather than incorporated in a review format to illustrate some emerging points or hypothesis in the field.

In sum, I am confused by the introduction of the manuscrit suggesting that new information will be provided while it is actually not the case; the content of the manuscript would better fit a regular review format. Again, this is my own opinion.

Author Response

Dear reviewer,

Upon the advice of the editor, we have addressed the minor concern about the “general scheme describing histone structure in complex with DNA” requested by reviewer number 2 (see new lines 125-135), and have not changed the format of the article.

Thank you for taking the time to review this article.

Warm regards,

Juan Ausió, on behalf of all the authors